# Sirt2 facilitates hepatic glucose uptake by deacetylating glucokinase regulatory protein

Hitoshi Watanabe[1], Yuka Inaba[1], Kumi Kimura[1], Michihiro Matsumoto[2], Shuichi Kaneko[3], Masato Kasuga[4] & Hiroshi Inoue[1]

Impaired hepatic glucose uptake (HGU) causes postprandial hyperglycemia in type 2 diabetes. Here, we show that diminished hepatic Sirt2 activity impairs HGU in obese diabetic mice. Hepatic Sirt2 overexpression increases HGU in high-fat diet (HFD)-fed obese diabetic mice and mitigates their impaired glucose tolerance. Hepatic Sirt2 knockdown in non-diabetic mice reduces HGU and causes impaired glucose tolerance. Sirt2 promotes glucose-dependent HGU by deacetylating K126 of glucokinase regulatory protein (GKRP). Glucokinase and GKRP glucose-dependent dissociation is necessary for HGU but is inhibited in hepatocytes derived from obese diabetic mice, depleted of Sirt2 or transfected with GKRP acetylation-mimicking mutants. GKRP deacetylation-mimicking mutants dissociate from glucokinase in a glucose concentration-dependent manner in obese diabetic mouse-derived hepatocytes and increase HGU and glucose tolerance in HFD-induced or db/db obese diabetic mice. We demonstrate that Sirt2-dependent GKRP deacetylation improves impaired HGU and suggest that it may be a therapeutic target for type 2 diabetes.

[1] Metabolism and Nutrition Research Unit, Institute for Frontier Science Initiative, Kanazawa University, Kanazawa, Ishikawa 920-8641, Japan. [2] Department of Molecular Metabolic Regulation, Diabetes Research Center, Research Institute, National Center for Global Health and Medicine, Tokyo 162-8655, Japan. [3] Department of System Biology, Kanazawa University Graduate School of Medical Sciences, Kanazawa, Ishikawa 920-8641, Japan. [4] National Center for Global Health and Medicine, Tokyo 162-8655, Japan. Correspondence and requests for materials should be addressed to H.I. (email: inoue-h@staff.kanazawa-u.ac.jp)

Postprandial hyperglycemia is closely associated with an increased risk of cardiovascular complications and lethality in type 2 diabetes[1–3]. Hepatic glucose uptake (HGU) accounts for one-third of food-derived exogenous glucose uptake[4] and impaired HGU in obesity and type 2 diabetes causes postprandial hyperglycemia[5,6]. Faulty hepatic glucokinase activation is reported to be responsible for the HGU impairment in type 2 diabetes[5,7], but the precise mechanism of the HGU and glucokinase activation impairment remains to be elucidated.

Glucokinase plays a prominent role in postprandial HGU[8]. Glucokinase catalyzes the phosphorylation of glucose to glucose-6-phosphate, the reverse of the reaction catalyzed by the

**Fig. 1** NAD+ restoration increases glucose uptake in steatotic hepatocytes. **a**, **b** NAD+ (**a**) and 2-deoxyglucose (2-DG) uptake (**b**) levels in primary hepatocytes derived from lean control or db/db mice with or without nicotinamide mononucleotide (NMN) ($n = 3$). **c** Schematic representation of NAD+ metabolism. **d** The protein level of glucokinase (GK) in primary hepatocytes derived from lean and db/db mice ($n = 3$). **e**, **f** NAD+ (**e**) and 2-DG uptake (**f**) levels in mouse primary hepatocytes treated with siNampt in the presence or absence of NMN ($n = 3$). **g**, **h** NAD + (**g**) and 2-DG uptake (**h**) levels in mouse primary hepatocytes treated with gallotannin (GTN) in the presence or absence of NMN ($n = 3$). **i** Glucose tolerance testing (1 g/kg) and the area under the curve (AUC) of the glucose tolerance test in normal chow (NC) mice and high-fat diet (HFD) mice treated with or without NMN during continuous dosing with somatostatin ($n = 5$). **j** Hyperinsulinemic-hyperglycemic clamping in HFD mice treated with or without NMN, performed as shown in the schematic. **k**–**m** 2-DG uptake levels in the liver, white adipose tissue (WAT), and skeletal muscle of HFD mice ($n = 6$) (**k**), hepatic *G6pc* and *Gck* mRNA expressions ($n = 6$) (**l**), and hepatic GK protein expression (**m**) during hyperinsulinemic-hyperglycemic clamping. *$P < 0.05$; one-way analysis of variance (ANOVA) with the Fisher's PLSD post-hoc test (**a**, **b**, **d**–**h**, **i** right) and Student's *t*-test (**k**, **l**), *$P < 0.05$, compared with NC, #$P < 0.05$, HFD vs. HFD with NMN in (**i**) left; one-way ANOVA with the Fisher's PLSD post -hoc test. Data in **d**, **m** are representative of at least three independent experiments. Error bars show the standard error of the mean (s.e.m.)

gluconeogenic enzyme glucose-6-phosphatase (G6Pase). HGU is based on the balance between glucokinase and G6Pase[9], but hepatic G6Pase activity is not significantly altered until 90 min postprandial[10]. The acute increase in hepatic glucokinase activity in response to the postprandial increase in blood glucose level is mainly induced by a post-translational mechanism dependent on glucokinase regulatory protein (GKRP)[8]. GKRP inhibits glucokinase activity by binding to the enzyme, and glucose dissociates

GKRP from glucokinase and thereby activates glucokinase[8,11]. Small-molecule glucokinase–GKRP disruptors that promote the dissociation of glucokinase–GKRP binding lower blood glucose levels in obese diabetic mice[12], indicating the importance of GKRP in maintaining blood glucose homeostasis. However, the role of GKRP in the HGU impairment in type 2 diabetes remains unclear.

Hepatic glucose metabolism derangement in type 2 diabetes is closely associated with depleted nicotinamide adenine dinucleo-

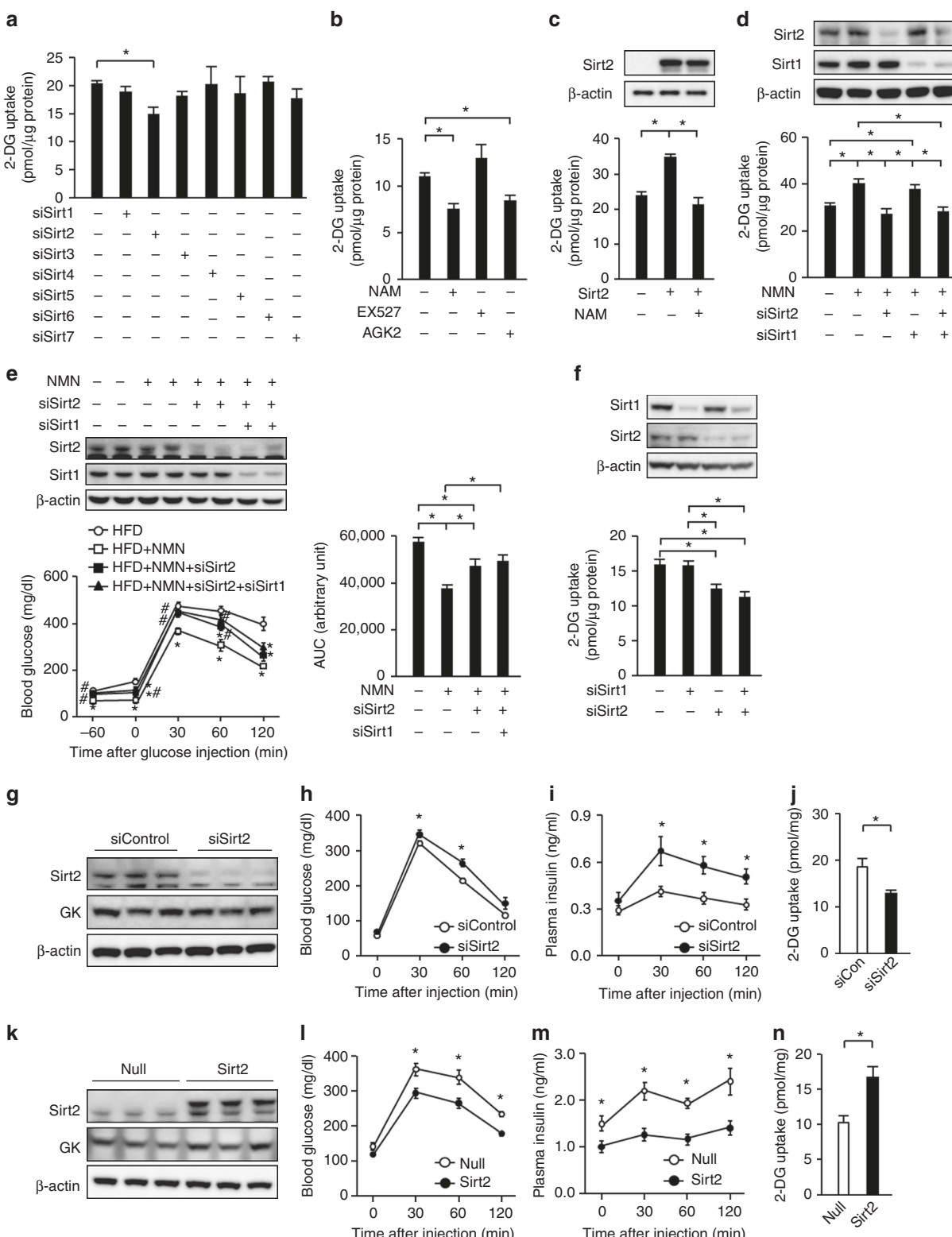

tide (NAD+), an energy balance indicator[13–15]. A high-fat diet (HFD) reduces hepatic NAD+ levels, and this NAD+ reduction in HFD-fed mice results from compromised NAD+ biosynthesis mediated by nicotinamide phosphoribosyltransferase (Nampt), the rate-limiting enzyme in NAD+ biosynthesis[16]. Furthermore, administration of the NAD+ precursor nicotinamide mononucleotide (NMN) restores the hepatic NAD+ level, which counteracts the impaired glucose tolerance caused by HFD intake[16]. Sirtuins, which are NAD+-dependent deacetylases, play a significant role in the effects of NAD+ on hepatic glucose metabolism. There are seven mammalian sirtuins (Sirt1 to 7), with Sirt1 and 3 involved in glucose and energy metabolism[15,17]. Sirt1, expressed in cell nuclei, deacetylates the transcriptional regulation of metabolism-related genes and regulates hepatic glucose metabolism[14,18]. Sirt3 deacetylates mitochondrial enzymes and promotes fatty acid oxidation and ketogenesis[19,20]. The function of Sirt2 partially overlaps with that of Sirt1 because both deacetylate the histone H4K16[21], but its role in hepatic glucose metabolism regulation is unclear. Sirt2 is mainly present in the cytosol and is involved in the regulation of cytoskeletal proteins[22]; in the liver, Sirt2 is reported to deacetylate keratin 8[23].

Here, we found that NAD+ depletion is involved in the HGU impairment of obese diabetic mice and that Sirt2 plays an important role in this process via GKRP deacetylation. Moreover, impeded Sirt2-dependent GKRP deacetylation reduced HGU and glucose tolerance, and Sirt2 overexpression or deacetylation-mimicking GKRP mutant ameliorated the HGU impairment and improved glucose tolerance in obese diabetic mice.

## Results

**NAD+ restoration increases HGU in obese diabetic mice.** Primary hepatocytes derived from leptin receptor-deficient obese diabetic db/db mice contained numerous lipid droplets, had lower NAD+ content, and displayed reduced 2-deoxyglucose (2-DG) uptake compared with their lean mouse-derived counterparts (Fig. 1a, b and Supplementary Fig. 1a). The NAD+ precursor NMN (Fig. 1c) restored the NAD+ levels in db/db mouse-derived primary hepatocytes and increased their 2-DG uptake (Fig. 1a, b). There was no clear change in glucokinase protein expression or in the expressions of *G6pc* and *Gck* genes, encoding G6Pase and glucokinase, respectively (Fig. 1d and Supplementary Fig. 1b). A reduction in NAD+ levels in lean mouse-derived primary hepatocytes via Nampt knockdown decreased the 2-DG uptake, despite the increase in *Gck* gene expression (Fig. 1e, f and Supplementary Fig. 1c). Restoration of the NAD+ content of Nampt knockdown hepatocytes with NMN reversed the 2-DG uptake impairment without changing the *Gck* gene expression (Fig. 1e, f and Supplementary Fig. 1c). Exposure to gallotannin (Fig. 1c), a nicotinamide mononucleotide adenylyltransferase (Nmnat) inhibitor that catalyzes NAD+ biosynthesis from NMN, reduced the

NAD+ quantity regardless of the presence of NMN (Fig. 1g), as reported previously[16]. In addition, 2-DG uptake was impaired; this impairment was not reversed by NMN (Fig. 1h). Gallotannin exposure reduced *G6pc* gene expression but did not have any clear effect on *Gck* gene expression (Supplementary Fig. 1d).

Obese diabetic HFD-fed mice had less hepatic NAD+ than mice fed a normal chow (NC) diet (Supplementary Fig. 2a). Intraperitoneal NMN-administered HFD-fed mice showed restored hepatic NAD+ levels, increased insulin secretion, and ameliorated impaired glucose tolerance (Supplementary Fig. 2a–c), as reported before[16]. We further studied the effect of NMN on the blood glucose level under the inhibition of endogenous insulin secretion via continuous administration of somatostatin (Supplementary Fig. 2d). The impaired glucose tolerance of the HFD-fed mice was improved by NMN administration, even with somatostatin administration, coincident with an increase in hepatic NAD+ levels (Fig. 1i and Supplementary Fig. 2e). Next, we investigated the effect of NMN administration on 2-DG uptake by the liver, white adipose tissue (WAT), and skeletal muscle in obese diabetic HFD-fed mice during hyperinsulinemic-hyperglycemic clamping (Fig. 1j). In the NMN-administered group, 2-DG uptake was augmented in the liver, although *G6pc* gene expression and glucokinase gene and protein expressions were unaffected (Fig. 1k–m). No clear changes were observed in the skeletal muscle and WAT (Fig. 1k).

**Sirt2 knockdown impedes NAD+-dependent HGU.** To elucidate the role of sirtuins in the HGU reduction after cellular NAD+ depletion, we knocked down Sirt1–7 in primary hepatocytes and investigated the effect on glucose uptake. Primary hepatocyte 2-DG uptake was only reduced by Sirt2 knockdown, without changes in *Gck* and *G6pc* gene expressions (Fig. 2a and Supplementary Fig. 3a). The sirtuin inhibitor nicotinamide and the Sirt2 inhibitor AGK2 reduced 2-DG uptake by primary hepatocytes, unlike the Sirt1 inhibitor EX527 (Fig. 2b). Sirt2 overexpression increased 2-DG uptake by db/db mouse-derived primary hepatocytes, but Sirt1 overexpression did not, and the Sirt2-induced increase in 2-DG uptake was inhibited by nicotinamide (Fig. 2c and Supplementary Fig. 3b). NMN increased the 2-DG uptake by db/db mouse-derived primary hepatocytes, but this effect was lost after Sirt2 knockdown, unlike Sirt1 knockdown (Fig. 2d). While NMN treatment ameliorated glucose intolerance in obese HFD-fed mice, Sirt2 knockdown partially negated the blood glucose-lowering effects of NMN in a glucose tolerance test when endogenous insulin secretion was inhibited by somatostatin administration (Fig. 2e and Supplementary Fig. 3c). In spite of the functional redundancy between Sirt1 and Sirt2[21], Sirt1 and Sirt2 double knockdown revealed no difference in 2-DG uptake compared with Sirt2 knockdown in lean mouse-derived hepatocytes or in NMN-treated db/db mouse-derived hepatocytes (Fig. 2d, f).

**Fig. 2** Sirt2 knockdown impedes NAD+-dependent hepatocyte glucose uptake. **a**, **b** Uptake of 2-deoxyglucose (2-DG) in primary hepatocytes treated with siSirt1–7 (**a**) and nicotinamide (NAM), Ex-527, or AGK2 (**b**) (n = 3). **c** Effect of adenovirus-mediated overexpression of Sirt2 on 2-DG uptake in primary hepatocytes derived from db/db mice in the presence or absence of NAM (n = 3). **d** Effect of Sirt2 and Sirt1 knockdown on 2-DG uptake in primary hepatocytes derived from db/db mice in the presence or absence of NMN (n = 4). **e** Glucose tolerance testing (1 g/kg) and the area under the curve (AUC) of the glucose tolerance test in HFD mice treated with NMN in the presence or absence of hepatic Sirt2 knockdown or Sirt2 and Sirt1 double knockdown during continuous dosing with somatostatin (n = 5). **f** Uptake of 2-DG in primary hepatocytes treated with siSirt1, siSirt2 or siSirt1 and siSirt2 (n = 3). **g–j** Effect of hepatic Sirt2 knockdown on hepatic protein expression of Sirt2 and glucokinase (GK) (**g**) and levels of blood glucose (n = 6) (**h**) and plasma insulin (n = 6) (**i**) and hepatic 2-DG uptake (n = 4) (**j**) after glucose administration (2 g/kg) in lean mice. **k–n** Effect of adenovirus-mediated hepatic Sirt2 overexpression on immunoblot analysis of Sirt2 and GK in the liver (**k**) and levels of blood glucose (n = 7) (**l**) and plasma insulin (n = 7) (**m**) and hepatic 2-DG uptake (n = 5) (**n**) after glucose administration (1 g/kg) in HFD mice. *P < 0.05; one-way ANOVA with the Fisher's PLSD post-hoc test (**a–d**, **e** right, **f**) and Student's t-test (**h–j**, **l–n**) *P < 0.05, compared with HFD, #P < 0.05, HFD+NMN vs. HFD+NMN with hepatic Sirt2 knockdown or Sirt2 and Sirt1 double knockdown in **e** left; one-way ANOVA with the Fisher's PLSD post-hoc test. Data in **g**, **k** are representative of at least three independent experiments. Error bars show s.e.m

Furthermore, hepatic Sirt1 knockdown had no additional amelioration of the NMN-dependent increase in glucose tolerance compared with hepatic Sirt2 knockdown (Fig. 2e and Supplementary Fig. 3c). These findings suggest that Sirt2 plays important roles in NAD+-dependent regulation of HGU.

We then investigated the role of Sirt2 in HGU and glucose homeostasis. Hepatic Sirt2 knockdown caused a slight increase in the blood glucose level and clear hyperinsulinemia under ad libitum feeding (Fig. 2g and Supplementary Fig. 3d, e). Increased blood glucose and plasma insulin levels were also observed in the glucose tolerance test (Fig. 2h, i). Hepatic 2-DG uptake diminished with the glucose load, but the uptake by WAT and the skeletal muscle was not different from that of the control (Fig. 2j and Supplementary Fig. 3f). Meanwhile, liver-specific Sirt2 overexpression by adenoviral vector in obese HFD-fed mice lowered the blood glucose and plasma insulin levels under ad

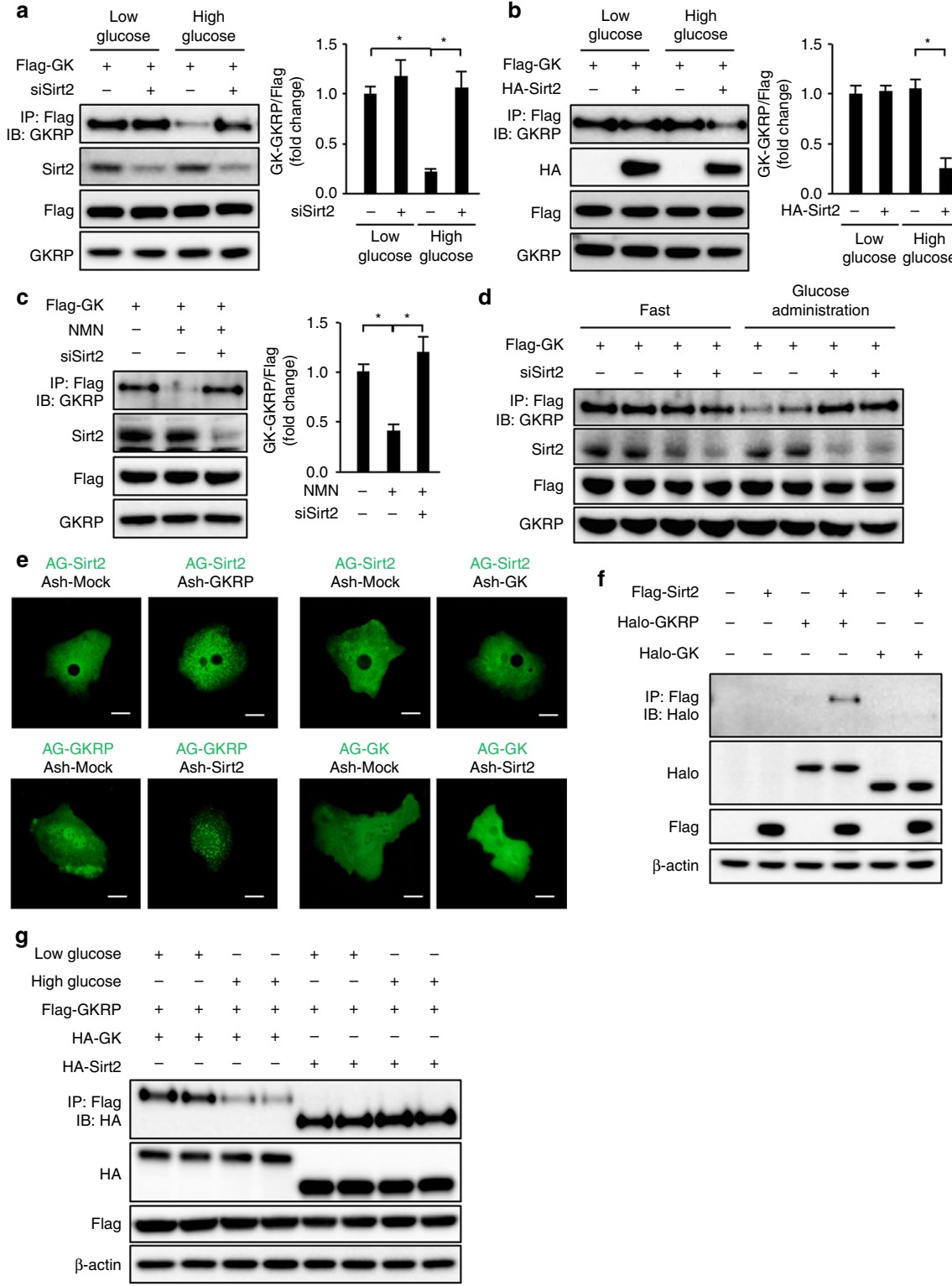

libitum feeding (Fig. 2k and Supplementary Fig. 3g, h). Their blood glucose and plasma insulin levels also decreased in the glucose tolerance test (Fig. 2l, m). There was no difference in WAT and skeletal muscle glucose uptake, but hepatic 2-DG uptake increased (Fig. 2n and Supplementary Fig. 3i). No clear change in hepatic *G6pc* gene expression was observed with hepatic Sirt2 knockdown or hepatic Sirt2 overexpression in obese HFD-fed mice (Supplementary Fig. 3j, k). The gene expression of *Gck* increased with hepatic Sirt2 knockdown despite the decrease in HGU, whereas the opposite effects were observed with hepatic Sirt2 overexpression (Supplementary Fig. 3j, k). These results suggest that Sirt2 operates independently from *Gck* and *G6pc* gene expressions to regulate HGU.

**Sirt2 facilitates the dissociation of glucokinase and GKRP.** To determine how Sirt2 regulates HGU, we investigated the effects of Sirt2 on the glucose-dependent dissociation of glucokinase and GKRP. While glucokinase and GKRP were bound in primary hepatocytes in low-glucose culture medium and dissociated under high-glucose conditions, Sirt2 knockdown inhibited the glucokinase–GKRP dissociation in high-glucose culture medium (Fig. 3a). In db/db mouse-derived primary hepatocytes, glucokinase and GKRP did not dissociate under high-glucose conditions, with Sirt2 overexpression restoring this dissociation (Fig. 3b). Treatment of db/db mouse-derived primary hepatocytes with NMN restored the glucokinase–GKRP binding dissociation under high-glucose conditions, but this effect was lost with Sirt2 knockdown (Fig. 3c). Glucose administration induced glucokinase–GKRP dissociation in the liver, but this dissociation was inhibited by hepatic Sirt2 knockdown (Fig. 3d).

To determine whether there is a direct effect of Sirt2 on glucokinase or GKRP, we investigated protein−protein interactions using fluorescent Azami Green (hAG) and Ash tags. When molecules possessing these tags bind directly to each other, hAG is concentrated locally and becomes visible as fluorescent foci. AG-Sirt2 was present in the cytoplasm in primary hepatocytes. Fluorescent foci appeared in the cytoplasm in the group transfected with both Sirt2 and GKRP, but no fluorescent foci were observed in the Sirt2- and glucokinase-transfected group (Fig. 3e). In HEK293 cells coexpressing Sirt2 and GKRP or glucokinase, Sirt2 and GKRP co-immunoprecipitated, unlike Sirt2 and glucokinase (Fig. 3f). In contrast, Sirt1 did not co-immunoprecipitate GKRP (Supplementary Fig. 4a). The Sirt2–GKRP co-immunoprecipitation remained unaltered by the glucose concentration in the culture medium in primary cultured hepatocytes, whereas the glucokinase–GKRP co-immunoprecipitation was blocked under high-glucose conditions (Fig. 3g). These results suggest that Sirt2 interacts directly with GKRP to regulate the dissociation of glucokinase and GKRP.

**Sirt2 deacetylates GKRP.** GKRP was acetylated in HEK293 cells, but the degree of acetylation was reduced by Sirt2 overexpression (Fig. 4a). This GKRP acetylation-reducing effect of Sirt2 overexpression was inhibited by nicotinamide and AGK2 but not by EX527 or the class I and II histone deacetylase inhibitor trichostatin A (Fig. 4a). GKPR acetylation in primary hepatocytes was enhanced by Sirt2 knockdown but was unaffected by the glucose concentration in the culture medium, as with NAD+ levels, even in the hepatocytes overexpressing GKRP (Fig. 4b and Supplementary Fig. 4b, c). GKRP acetylation was enhanced in db/db mouse-derived primary hepatocytes but was reduced by Sirt2 overexpression (Fig. 4c). NMN treatment also reduced GKRP acetylation in db/db mouse-derived primary hepatocytes, and this effect was blocked by Sirt2 knockdown (Fig. 4d). Furthermore, in livers of obese HFD-fed mice, forced hepatic Sirt2 expression using an adenoviral vector reduced GKRP acetylation (Fig. 4e). These findings suggest that Sirt2 deacetylates GKRP in a NAD+-dependent manner.

We evaluated hepatic Sirt2 activity by measuring deacetylation of keratin 8, a known hepatic target protein of Sirt2-dependent deacetylation[23], in addition to GKRP deacetylation. Hepatic Sirt2 knockdown resulted in enhanced acetylation of both keratin 8 and GKRP in mouse liver (Fig. 4f). GKRP acetylation was increased in the livers of obese HFD-fed mice but was reduced by NMN, coincident with keratin 8 acetylation (Fig. 4g). The acetylation of both Sirt2 and keratin 8 was unaffected by glucose administration (Fig. 4h). While acetylation of both hepatic keratin 8 and GKRP was much weaker in lean mice than in obese HFD-fed mice under ad libitum feeding, fasting diminished acetylation, which then slightly increased 6 h after refeeding (Supplementary Fig. 4d).

**GKRP deacetylation facilitates glucokinase−GKRP dissociation.** Next, we investigated the acetylation sites of GKRP. DYKDDDDK-tagged GKRP expressed in HEK293 cells was analyzed by liquid chromatography tandem mass spectrometry (LC-MS/MS). We identified K43, K126, K165, K235, K267, K279, K312, and K327 as GKRP acetylation site candidates (Supplementary Fig. 5a−h). A deacetylation-mimicking lysine (K)-to-arginine (R) mutant and an acetylation-mimicking lysine-to-glutamine (Q) mutant were created for each candidate to determine the Sirt2-dependent deacetylation site. Only K126R and K126Q were not deacetylated by Sirt2 (Fig. 5a, b). Furthermore, K126R mutant overexpression suppressed the GKRP acetylation that was enhanced in db/db mouse-derived primary hepatocytes, suggesting that K126 of GKRP is the site of the acetylation in the hepatocytes of diabetic obese mice (Fig. 5c). The K126R mutant behaved like wild-type GKRP in primary hepatocytes, that is, binding to and dissociating from glucokinase

**Fig. 3** Sirt2 regulates glucokinase and GKRP dissociation. **a–c** Immunoblot analysis (IB) with the indicated antibodies of total cell lysate and precipitates after immunoprecipitation (IP) with anti-DYKDDDDK (Flag) tag antibodies from Sirt2-knockdown primary hepatocytes transfected with adenovirus vector encoding Flag-tagged glucokinase (Flag-GK) cultured in low- or high-glucose medium (**a**), from db/db mouse-derived hepatocytes transfected with adenovirus vectors encoding Flag-GK and HA-tagged Sirt2 (HA-Sirt2) cultured in low- or high-glucose medium (**b**), and from db/db mouse-derived primary hepatocytes transfected with adenovirus vectors encoding Flag-GK with and without Sirt2 knockdown cultured in high-glucose medium with and without NMN (**c**). GKRP levels coimmunoprecipitated with Flag-GK were normalized to Flag-GK expression and are shown in the right graph of each figure (*n* = 3 each). **d** Effect of Sirt2 knockdown on immunoprecipitation and immunoblot analysis of GK−GKRP binding in the liver from mice transfected with adenovirus vectors encoding Flag-GK 30 min after administration of glucose (4 g/kg) after 16-h fasting. **e** The interaction between Sirt2 and GKRP or glucokinase in primary hepatocytes detected by a fluorescent-based system for detecting protein−protein interactions using hAG-tagged Sirt2, hAG-tagged GKRP, hAG-tagged GK, Ash-Sirt2, Ash-GKRP, and Ash-GK. Scale bars indicate 20 μm. **f** Immunoprecipitation and immunoblot analysis of the interaction between Sirt2 and GKRP or Sirt2 and GK in HEK293 cells expressing Flag-tagged Sirt2, Halo-tagged GKRP, or Halo-tagged GK. **g** Immunoprecipitation and immunoblot analysis of GK−GKRP or Sirt2−GKRP binding in mouse primary hepatocytes transfected with adenovirus encoding Flag-GKRP and HA-GK or HA-Sirt2 in low- or high-glucose medium. Data are normalized to control (**a–c**). *$P < 0.05$; one-way ANOVA with the Fisher's PLSD post-hoc test (**a–c**). All data are representative of at least three independent experiments. Error bars show s.e.m.

under low- and high-glucose conditions, respectively (Fig. 5d). Sirt2 knockdown or sirtuin inhibition by nicotinamide impeded the glucose-dependent dissociation of wild-type GKRP from glucokinase; with K126R mutant, dissociation from glucokinase was not impeded by Sirt2 knockdown or nicotinamide under high-glucose conditions (Fig. 5d, e). In wild-type GKRP-overexpressing hepatocytes, Sirt2 knockdown and nicotinamide treatment reduced 2-DG uptake. In K126R mutant-expressing hepatocytes, the 2-DG uptake-impeding effect of these

conditions was mitigated (Fig. 5f, g). In db/db mouse-derived primary hepatocytes, wild-type GKRP was bound to glucokinase regardless of the glucose concentration in the culture medium, whereas K126R mutant dissociated from glucokinase in a glucose concentration-dependent manner (Fig. 5h). K126R mutant increased 2-DG uptake in db/db mouse-derived primary hepatocytes without changes in NAD+ levels (Fig. 5i, j). In contrast, K126Q mutant bound strongly to glucokinase under high-glucose conditions (Fig. 5k, l) and decreased 2-DG uptake

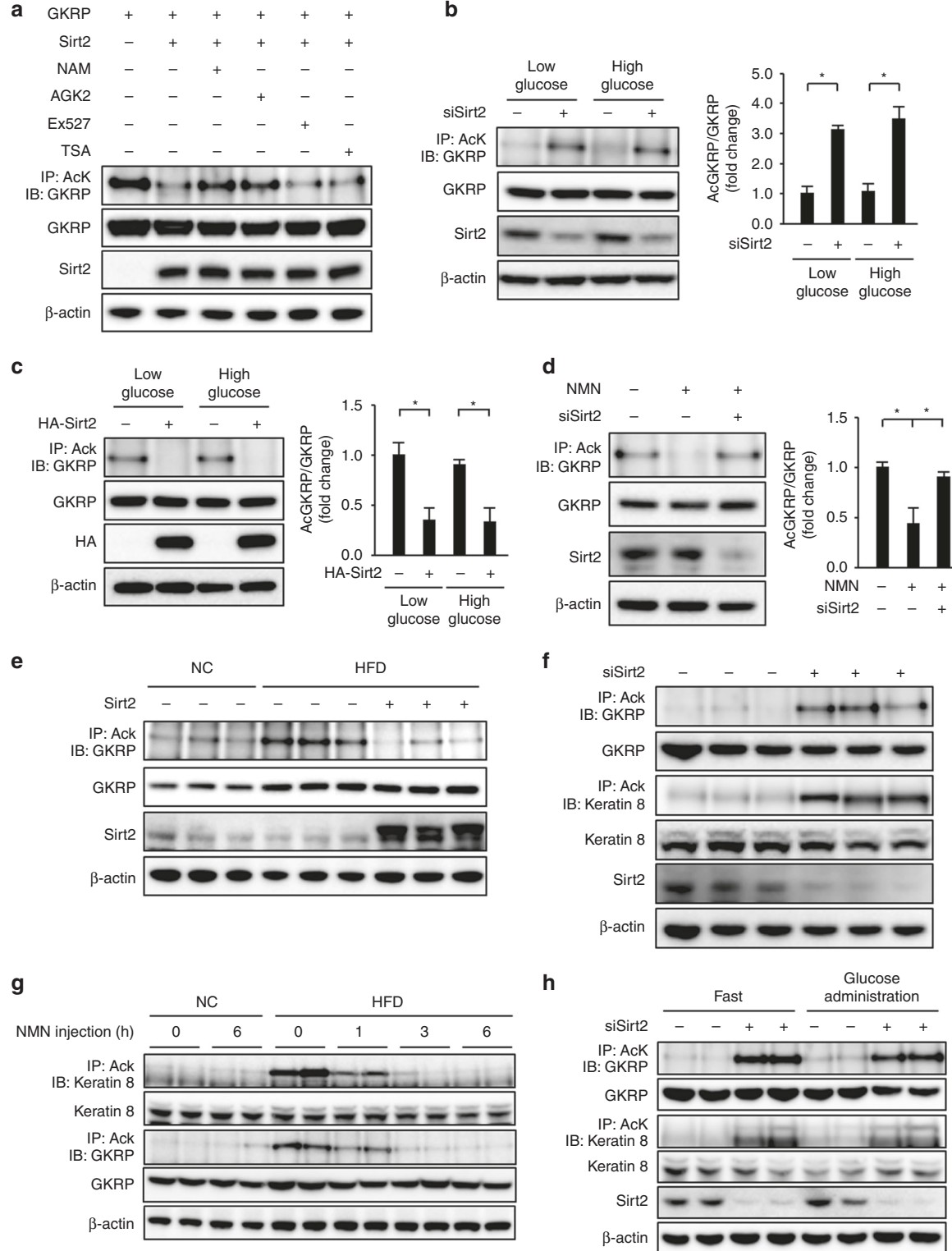

(Fig. 5m, n) to a similar extent to Sirt2 knockdown and nicotinamide treatment. Transfection with either mutant gene did not have any clear effect on *Gck* or *G6pc* gene expressions (Supplementary Fig. 6a, b).

Hepatic Sirt2 knockdown diminished hepatic 2-DG uptake in mice transduced with adenoviral vector encoding wild-type GKRP, causing slightly impaired glucose tolerance and elevated plasma insulin levels (Fig. 6a–c and Supplementary Fig. 7a). K126R mutant transduction in hepatic Sirt2 knockdown mice mitigated all of these conditions (Fig. 6a–c and Supplementary Fig. 7a). In obese HFD-fed mice, intraperitoneal NMN increased the plasma insulin value, improved glucose tolerance, and increased hepatic 2-DG uptake. Hepatic K126Q overexpression diminished these effects of NMN, although there was no alteration in the plasma insulin-elevating effect (Fig. 6d–f and Supplementary Fig. 7b). Hepatic *Gck* and *G6pc* gene expressions were unaffected by transduction with K126R or K126Q mutant (Supplementary Fig. 7c, d).

**GKRP deacetylation improves HGU in obese diabetic mice.** Hepatic K126Q-overexpressing mice displayed elevated blood glucose and plasma insulin values when fed ad libitum (Fig. 7a, b and Supplementary Fig. 8a). No changes in hepatic NAD+ levels or the gene expression of *Gck* and *G6pc* were observed, whereas hepatic 2-DG uptake following a glucose load was reduced and blood glucose and plasma insulin values were increased (Fig. 7c–e and Supplementary Fig. 8b–d). Hepatic K126R overexpression in obese HFD-fed mice did not alter the blood glucose value but reduced the plasma insulin values, suggesting that it improved the insulin sensitivity (Fig. 7f, g and Supplementary Fig. 8e). Their hepatic 2-DG uptake was augmented, followed by lowered blood glucose and plasma insulin values in the glucose tolerance test (Fig. 7h–j and Supplementary Fig. 8f). There was no change in the hepatic NAD+ levels and the gene expressions of *Gck* and *G6pc* (Supplementary Fig. 8g, h). In db/db mice, blood glucose and plasma insulin values diminished as a result of hepatic K126R overexpression, under both ad libitum feeding and glucose tolerance test conditions (Fig. 7k–n). Hepatic 2-DG uptake in db/db mice increased but no changes in hepatic NAD+ levels and the gene expression of glucokinase and *G6pc* were observed as a result of K126R overexpression (Fig. 7o and Supplementary Fig. 8i–k). While wild-type GKRP overexpressed in the liver was acetylated, that of K126R mutant was not acetylated in db/db mice (Fig. 7p). These results suggest that K126 acetylation of GKRP has a significant effect on HGU and glucose tolerance.

## Discussion
Impaired HGU results in the postprandial hyperglycemia associated with type 2 diabetes[5,6], but its molecular mechanism is unclear. We found that HGU impairment in obese diabetic mice results in part from a decrease in hepatic NAD+-dependent Sirt2

activity and a defect in Sirt2-dependent deacetylation of K126 of GKRP. NAD+ levels in the liver are decreased in obese diabetic mice, as well as in the skeletal muscles, adipose tissues, and pancreatic β cells[16]. NMN administration can restore the NAD+ levels in these organs, enhancing glucose tolerance through increased insulin sensitivity in female mice and insulin secretion in male mice[16]. Here, we confirmed that insulin secretion and glucose tolerance were enhanced by NMN administration. In addition, we found that the NMN-induced NAD+ content increase led to enhanced glucose tolerance via an insulin-independent increase in HGU, as well as an insulin-mediated manner. Indeed, NMN administration reduced the blood glucose value during glucose tolerance testing and increased hepatic 2-DG uptake under a somatostatin-administered state during which endogenous insulin secretion is inhibited. In primary hepatocytes, NMN restored the Nampt knockdown-induced decrease in HGU but could not restore the reduced HGU when NAD+ biosynthesis from NMN was inhibited. This shows that the HGU-enhancing effect of NMN depends on hepatocyte NAD+ content.

Impaired 2-DG uptake in db/db mouse-derived hepatocytes was mitigated by NMN, but this effect was lost after Sirt2 knockdown, indicating the involvement of Sirt2 in hepatic NAD+-dependent regulation of HGU. In lean mouse-derived hepatocytes and lean mouse livers, Sirt2 knockdown reduced 2-DG uptake, suggesting that Sirt2 is essential for maintaining physiological HGU and postprandial glucose homeostasis. In contrast, overexpression of Sirt2 increased hepatic 2-DG uptake in db/db mouse-derived hepatocytes and HFD-fed mouse livers, indicating the importance of Sirt2 in the glucose uptake impairment found in obese diabetic livers. In our study, the acetylation of keratin 8, which is deacetylated by Sirt2, was enhanced in the liver of HFD-fed mice, which is in line with previous reports that its acetylation is enhanced in obese diabetic leptin-deficient ob/ob mice[23]. However, keratin 8 acetylation remained unchanged after glucose administration, and its change after refeeding in lean mice remained minimal. If the hepatic levels of NAD+ and of keratin 8 and GKRP deacetylation reflect hepatic Sirt2 activity, it is plausible that a decrease in Sirt2 activity in the obese diabetic livers results in compromised regulation of HGU and glucokinase translocation by postprandial hyperglycemia. While the function of Sirt1 partially overlaps that of Sirt2, neither Sirt1 knockdown in lean mouse-derived hepatocytes nor its overexpression in db/db mouse-derived hepatocytes affected 2-DG uptake. Indeed, Sirt1 did not bind to GKRP, whereas Sirt2 directly bound to GKRP to regulate HGU.

Because HGU is regulated by both glucose and insulin[4], the insulin-independent glucose tolerance-enhancing effect of NAD+/Sirt2 may be in part responsible for the glucose-dependent regulation of HGU, which also involves glucokinase and GKRP. In studies using co-immunoprecipitation and hAG/Ash tags, GKRP was suggested to be the direct target protein of Sirt2. We then found that Sirt2 knockdown and overexpression resulted in increased and decreased GKRP acetylation, respectively, in

**Fig. 4** Sirt2 deacetylates GKRP. **a**–**d** Immunoblot analysis (IB) with the indicated antibodies of total cell lysate and precipitates after immunoprecipitation (IP) with anti-acetylated lysine (AcK) antibodies from HEK293 cells expressing Flag-tagged GKRP and Flag-tagged Sirt2 with and without nicotinamide (NAM), AGK2, Ex527, or trichostatin A (TSA) (**a**), from Sirt2 knockdown primary hepatocytes cultured in low- or high-glucose medium (**b**), from db/db mouse-derived hepatocytes with and without adenovirus-mediated overexpression of Sirt2 cultured in low- or high-glucose medium (**c**), and from db/db mouse-derived hepatocytes with NMN treatment and with and without Sirt2 knockdown cultured in high-glucose medium (**d**). The quantification of acetylated GKRP levels is normalized to GKRP expression in the right graph (**b**–**d**, $n = 3$ each). **e**, **f** Acetylated GKRP levels in the liver of mice fed normal chow (NC) or a high-fat diet (HFD) with adenovirus-mediated hepatic overexpression of Sirt2 after 16-h fasting (**e**) and acetylated GKRP and keratin 8 levels in the liver of Sirt2 knockdown lean mice after 16-h fasting (**f**). **g**, **h** Successive comparison of lysine acetylation levels of keratin 8 and GKRP in the liver of NC- or HFD-fed mice after NMN injection after 16-h fasting (**g**) and in the liver of NC-fed mice treated with siSirt2 at 30 min after administration of glucose (4 g/kg) after 16-h fasting (**h**). *$P < 0.05$; one-way ANOVA with the Fisher's PLSD post-hoc test (**b**–**d**). All data are representative of at least three independent experiments. Error bars show s.e.m

primary hepatocytes, suggesting that Sirt2 deacetylates GKRP. Hepatic *Gck* gene expression was enhanced after hepatic Sirt2 knockdown and was reduced after hepatic Sirt2 overexpression. These changes might result in underestimation of the effects of Sirt2 on HGU regulation. Because Sirt2 knockdown had no effect on the *Gck* gene expression in primary hepatocytes, these *Gck*

gene expression changes under Sirt2 knockdown and over-expression conditions may be caused by the change in the plasma insulin levels, which correlated with insulin sensitivity.

Mass spectrometry and mutant GKRP studies revealed that K126 of GKRP is the target deacetylation site of Sirt2. GKRP acetylation is advanced in the liver of HFD-fed mice, whereas

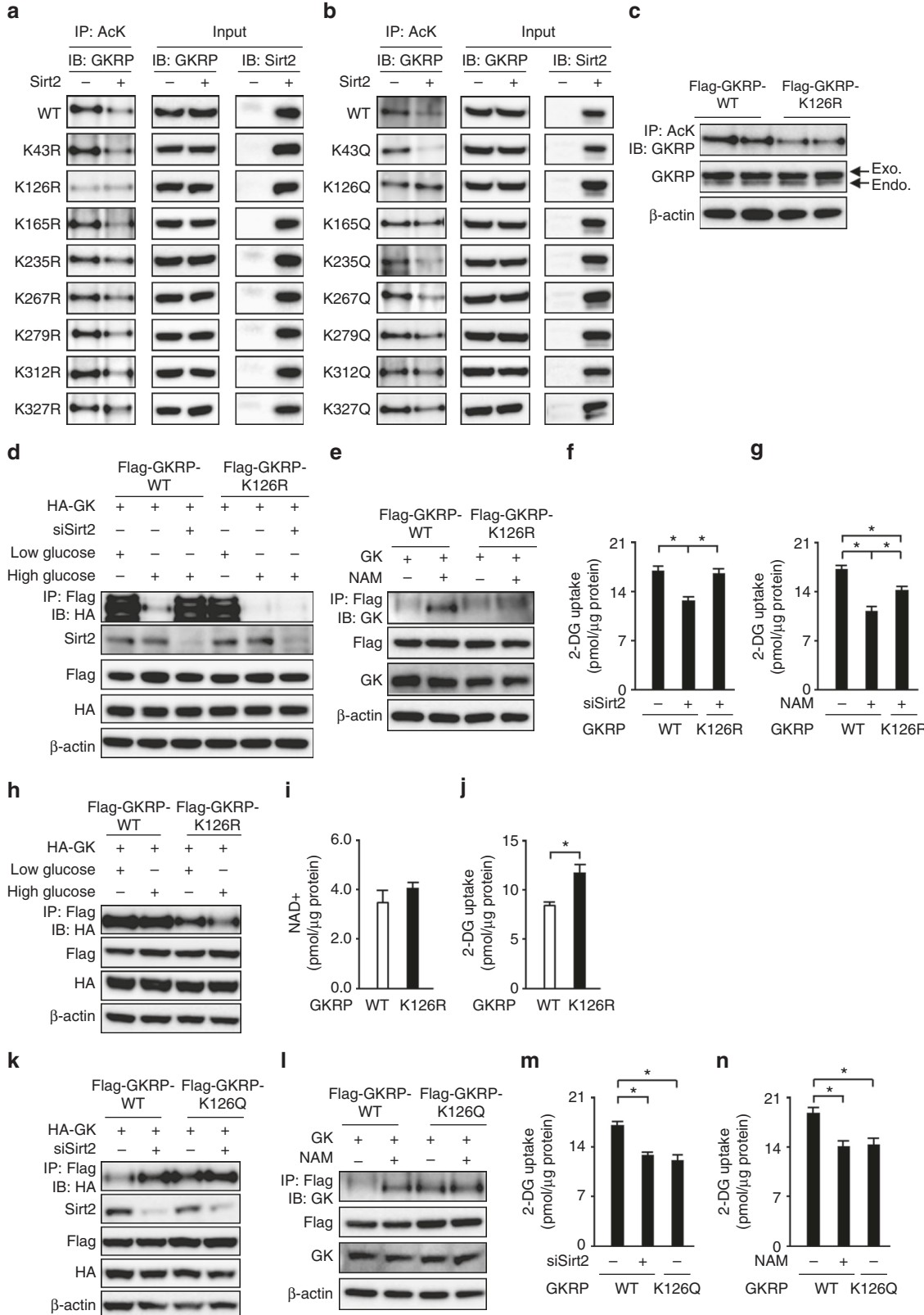

GKRP acetylation is alleviated in db/db mouse-derived hepatocytes and liver with hepatic K126R overexpression. This indicates that K126 undergoes enhanced acetylation following decreased Sirt2 activity in obese diabetic liver. GKRP acetylation still showed little change after glucose administration and a minimal increase after refeeding in lean mice, but Sirt2 knockdown resulted in elevated GKRP acetylation in lean mice. These findings reveal that Sirt2 activity suffices to deacetylate GKRP in the livers of lean mice, even under fasting and refeeding conditions. GKRP acetylation involves not only deacetylase, but also acetyltransferase. The detailed mechanism underlying the slight change in GKRP acetylation after refeeding is unclear but it may involve acetyltransferase. A study in HeLa cells identified p300 as the acetyltransferase responsible for K5 acetylation of GKRP[24], but further investigation is required to identify the mechanism of K126 acetylation found in our study.

Glucokinase is bound to GKRP under low-glucose conditions, resulting in its inactivation. Glucose induces a structural rearrangement of glucokinase, which is followed by glucokinase–GKRP dissociation and glucokinase activation[25]. Indeed, co-immunoprecipitation of glucokinase and GKRP decreased under high-glucose conditions in lean mouse-derived hepatocytes. Sirt2 appears to function as a NAD+-dependent rheostat that determines the extent of the interaction between GK and GKRP via GKRP deacetylation under a high-glucose condition. Actually, acetylation-mimicking GKRP mutant (K126Q) could not dissociate from glucokinase under high-glucose conditions. However, as deacetylation-mimicking GKRP mutant (K126R) can bind to glucokinase under high-glucose conditions, deacetylation of GKRP itself did not induce its dissociation from glucokinase. We also found that Sirt2 binds to GKRP and that this association in lean mouse-derived hepatocytes was unaffected by glucose levels in the cell medium. These findings may indicate that Sirt2 binds to and deacetylates GKRP to prevent GKRP from binding to and inhibiting glucokinase in lean mouse liver, where NAD+ levels are sufficient for Sirt2 to deacetylate GKRP even under high-glucose conditions; conversely, compromised Sirt2 activity caused by NAD+ reduction in obese-mouse liver fails to deacetylate GKRP, resulting in GKRP binding to and inhibiting glucokinase. Elucidation of how K126 acetylation prevents the glucose-dependent glucokinase–GKRP dissociation, such as via structural changes in their complex, remains a future challenge.

In vivo studies indicate that Sirt2-dependent GKRP K126 deacetylation plays an important role in the pathogenesis of HGU and glucose tolerance impairments in obese diabetic livers. Because an acetylation-mimicking mutation of GKRP (K126Q) inhibited glucose-dependent glucokinase–GKRP dissociation and diminished 2-DG uptake in mouse-derived hepatocytes, hepatic

acetylation-mimicking GKRP (K126Q)-expressing mice displayed reduced hepatic 2-DG uptake and impaired glucose tolerance and resistance to NMN-dependent amelioration of the impeded hepatic 2-DG uptake and glucose intolerance in obese HFD-fed mice. On the other hand, because the glucose-dependent glucokinase–GKRP dissociation was restored and 2-DG uptake was augmented by deacetylation-mimicking GKRP mutant (K126R) overexpression in db/db mouse-derived hepatocytes, deacetylation-mimicking GKRP (K126R) transduction in the livers of HFD-fed mice and db/db mice increased hepatic 2-DG uptake and improved glucose tolerance and insulin resistance, even though their hepatic NAD+ levels were unaffected. Furthermore, hepatic K126R transduction overcame hepatic Sirt2 knockdown-induced impaired hepatic 2-DG uptake and glucose intolerance.

This study revealed that NAD+/Sirt2 are important in HGU regulation and that Sirt2 permits glucose-dependent dissociation of glucokinase from GKRP by deacetylating K126 of GKRP. In HFD-fed obese diabetic mice and db/db mice, Sirt2 activity diminishes with the decrease in hepatic NAD+ content, which enhances GKRP acetylation and impairs HGU. Moreover, we found that K126 deacetylation of GKRP alleviates the impaired glucose tolerance and insulin resistance in obese diabetic HFD-fed mice and db/db mice. These findings suggest that NAD+/Sirt2-dependent GKRP deacetylation regulation plays an important role in HGU control and that this regulation is a novel therapeutic target in type 2 diabetes and obesity and is responsible for HGU impairment.

## Methods

**Cell culture.** Primary hepatocytes were isolated from 8–12-week-old male C57BL/6J or db/db mice and cultured using a previously described method[26]. The liver from these anesthetized mice was perfused at a rate of 4.5 ml/min for the first 3 min with Hank's balanced salt solution (HBSS) (Wako, Saitama, Japan) containing 10 mM Hepes-NaOH (pH 7.4) and then for 17 min with HBSS containing collagenase type I (0.3 mg/ml) (Worthington, Lakewood, NY) and Protease Inhibitor Cocktail Complete-EDTA free (one tablet per 50 ml) (Roche, Basel, Switzerland). Hepatocytes from C57BL/6J were purified by density gradient centrifugation with Percoll (Sigma, St. Louis, MO), seeded at a density of $5.0 \times 10^4$ cells/cm$^2$ and incubated in William's Medium E (Life Technologies, Carlsbad, CA) supplemented with 10% fetal bovine serum (FBS). For Sirt1–7 and Nampt in vitro knockdown models, hepatocytes were transfected with siRNA using Lipofectamine RNAiMAX (Life Technologies). In the in vitro 2-DG uptake assay, after primary hepatocytes were cultured in high-glucose FBS-free Dulbecco's modified Eagle medium (DMEM) (Wako) with Sirt activator NMN (100 μM) (Sigma), Nmnat inhibitor gallotannin (100 μM) (Santa Cruz Biotechnology, Santa Cruz, CA), nicotinamide (20 μM) (Sigma), Ex-527 (10 μM) (Tocris Bioscience, Ellisville, MO), or AGK2 (10 μM) (Santa Cruz Biotechnology) for 6 h, hepatocytes were incubated in PBS containing 2-DG (25 mM) (Sigma) for 3 min. In immunoprecipitation and immunoblot analysis of glucokinase–GKRP binding and the acetylated GKRP level, hepatocytes were incubated in FBS-free DMEM with low (5 mM) or high (25 mM) glucose concentrations with NMN (100 μM) or nicotinamide (20 μM) for 6 h. HEK 293 cells,

**Fig. 5** Deacetylation of K126 on GKRP by Sirt2 facilitates GK–GKRP dissociation and hepatocyte glucose uptake. **a**, **b** Comparison of the acetylation levels of wild-type (WT) GKRP, eight GKRP Lys-to-Arg mutants (**a**), and eight GKRP Lys-to-Gln mutants (**b**) in HEK293 cells expressing Flag-tagged Sirt2 or not. **c** The acetylation levels of GKRP in primary hepatocytes derived from db/db mice transfected with adenovirus encoding Flag-tagged GKRP-wild-type (Flag-GKRP-WT) or Flag-tagged GKRP-K126R (Flag-GKRP-K126R). Endo. endogenous GKRP, Exo. exogenous GKRP. **d** Effect of Sirt2 knockdown on glucokinase (GK)–GKRP binding in primary hepatocytes transfected with adenovirus encoding HA-tagged GK (HA-GK) and Flag-GKRP-WT or Flag-GKRP-K126R in low- and high-glucose medium. **e** Effect of nicotinamide (NAM) on GK–GKRP binding in mouse primary hepatocytes transfected with adenovirus encoding GK and Flag-GKRP-WT or Flag-GKRP-K126R in high-glucose medium. **f**, **g** Effect of Sirt2 knockdown (**f**) or NAM (**g**) on 2-deoxyglucose (2-DG) uptake in mouse primary hepatocytes transfected with adenovirus vector encoding Flag-GKRP-WT or Flag-GKRP-K126R ($n = 4$). **h** Immunoprecipitation and immunoblot analysis of GK–GKRP binding in db/db mouse primary hepatocytes transfected with adenovirus encoding HA-GK and Flag-GKRP-WT or Flag-GKRP-K126R in low- and high-glucose medium. **i**, **j** NAD+ (**i**) and 2-DG uptake (**j**) levels in db/db mouse primary hepatocytes transfected with adenovirus vector encoding Flag-GKRP-WT and Flag-GKRP-K126R ($n = 5$). **k**, **l** Effect of Sirt2 knockdown (**k**) and NAM (**l**) on GK–GKRP binding in primary hepatocytes transfected with adenovirus vectors encoding HA-GK or GK and Flag-GKRP-WT or Flag-tagged GKRP-K126Q (Flag-GKRP-K126Q) in high-glucose medium. **m**, **n** Effect of Sirt2 knockdown (**m**) or NAM (**n**) on 2-DG uptake in primary hepatocytes transfected with adenovirus vectors encoding Flag-GKRP-WT or Flag-GKRP-K126Q ($n = 4$). *$P < 0.05$; one-way ANOVA with the Fisher's PLSD post-hoc test (**f**, **g**, **m**, **n**) and Student's $t$-test (**i**, **j**). Data in **a–e**, **h**, **k**, **l** are representative of at least three independent experiments. Error bars show s.e.m

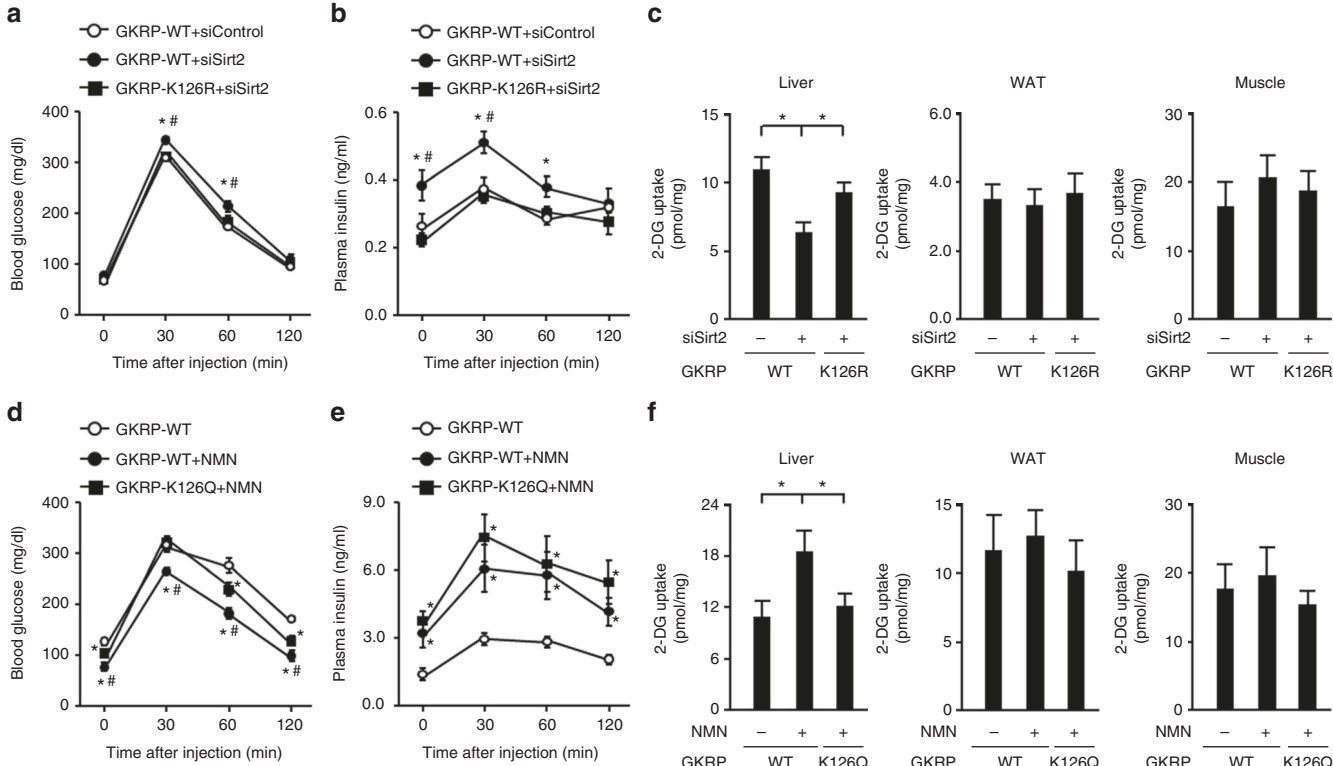

**Fig. 6** Deacetylation of K126 on GKRP plays an essential role in NAD+ /Sirt2-dependent hepatic glucose uptake. **a–c** Effect of adenovirus-mediated hepatic overexpression of GKRP-K126R or GKRP-wild-type (GKRP-WT) in hepatic Sirt2 knockdown mice on the levels of blood glucose ($n = 5$) (**a**), plasma insulin ($n = 5$) (**b**), and 2-deoxyglucose (2-DG) uptake in the liver, WAT, and skeletal muscle ($n = 4$) (**c**) after glucose administration (2 g/kg). **d–f** Effect of adenovirus-mediated hepatic overexpression of GKRP-K126Q or GKRP-WT in high-fat diet (HFD)-fed mice administered NMN on the levels of blood glucose ($n = 5$) (**d**), plasma insulin ($n = 5$) (**e**), and 2-DG uptake in the liver, WAT, and skeletal muscle ($n = 4$) (**f**) during glucose administration (1 g/kg). *$P < 0.05$; one-way ANOVA with the Fisher's PLSD post-hoc test (**c**, **f**), *$P < 0.05$, vs. WT in (**a**, **b**, **d**, **e**), #$P < 0.05$, WT+siSirt2 vs. K126R+siSirt2 (**a**, **b**) or WT+NMN vs. K126Q+NMN (**d**, **e**); one-way ANOVA with the Fisher's PLSD post-hoc test. Error bars show s.e.m

human embryonic kidney cells, were obtained from Cell Biolabs (San Diego, CA). HEK 293 cells were cultured in DMEM with 10% FBS and transfected with each plasmid for 24 h using FuGENE 6 transfection reagent (Promega, Madison, WI). HEK 293 cells were stimulated with NAM (20 μM), Ex-527 (10 μM), AGK2 (10 μM), and TSA (1 μM) for 6 h. The cell lines were regularly tested for mycoplasma contamination.

**Biochemical analysis**. NAD+ and 2-DG levels in primary hepatocytes and tissue samples were measured using a NAD/NADH quantification kit (BioVision, Mountain View, CA) and a 2-DG Uptake Measurement Kit (Cosmo Bio, Tokyo, Japan), respectively.

**Animals**. All animal experiments were approved by the Animal Ethics Committee of Kanazawa University and were performed according to the Animal Ethics Committee's guidelines for the care and use of laboratory animals at Kanazawa University. Eight-week-old male C57BL/6J and db/db mice were obtained from Japan SLC (Shizuoka, Japan) and maintained in a temperature-controlled environment with a 12-h light/dark cycle with free access to food and water. An HFD (61% of energy from fat) (D12492; Research Diet, New Brunswick, NJ) was fed to 8-week-old mice for 12 weeks. In NMN experiments, mice were intraperitoneally injected with NMN (500 mg/kg body weight) 6 h before each analysis. Liver-specific knockdown mice were generated by siRNA as described previously[27]. For the in vivo 2-DG uptake assay, except in hyperinsulinemic-hyperglycemic clamp experiments, 2-DG (200 μmol/kg) was injected into the tail vein 10 min after glucose administration (2 g/kg body weight for fasted lean mice; 1 g/kg body weight for fasted HFD mice). Liver, epididymal WAT, and soleus muscle samples were collected 10 min after 2-DG injection. For in vivo immunoprecipitation and immunoblot analysis of the GK–GKRP binding assay, livers were obtained from mice 30 min after glucose administration (4 g/kg) after 16-h fasting. Cages of mice were allocated to experimental groups by random draw. The investigator was not blinded to the group allocation.

**Glucose tolerance testing**. For intraperitoneal glucose tolerance tests, mice were intraperitoneally administered 1 g/kg body weight glucose for HFD-fed mice and 2 g/kg body weight glucose for NC-fed mice, with or without continuous intravenous administration of somatostatin (3 μg/kg/min) (Sigma) through a jugular catheter

from 1 h before glucose administration. Blood glucose levels were measured using a GLUCOCARD G+ Meter (Arkray, Kyoto, Japan) and a commercial colorimetric kit (Wako). Plasma insulin levels were measured using a Mouse Insulin ELISA Kit (Shibayagi, Gunma, Japan).

**Hyperinsulinemic-hyperglycemic clamping**. Hyperinsulinemic-hyperglycemic clamp studies were performed 4–7 days after canalization as described previously with modifications[28]. Hyperinsulinemic-hyperglycemic clamping was performed in awake and unrestrained overnight-fasted mice. During the clamp study, insulin (1.25 mU/kg/min) (Eli Lilly, Indianapolis, IN) and somatostatin (3 μg/kg/min) were infused through the jugular catheter with a variable amount of 40% glucose solution to maintain the blood glucose level at 280 ± 30 mg/dl. After 50 min, 2-DG (200 μmol/kg) was intravenously administered for 1 min and, 10 min after 2-DG administration, liver, WAT, and soleus muscle samples were obtained for a 2-DG uptake assay.

**Quantitative PCR**. Quantitative PCR was performed as previously described[29]. Total RNA was extracted from frozen tissue or cell samples using the SV Total RNA Isolation System (Promega). cDNA was synthesized from total RNA with the PrimeScript RT reagent kit (Takara Bio, Otsu, Japan). Quantitative PCR was performed by using the SYBR Select Master Mix kit (Life Technologies) with 36B4 as the control gene. The primer sequences used in this study are available upon request (36B4, Gck [which encodes glucokinase] and G6pc [which encodes G6Pase]).

**Western blotting and immunoprecipitation**. Western blotting was performed as previously described[30]. Liver tissues and cultured cells were homogenized in ice-cold CelLytic MT Cell Lysis Reagent (Sigma) with protease inhibitors. Immunoprecipitation was performed using Acetylated K antibodies conjugated to Dynabeads Protein G (Life Technologies) or Flag tag antibody conjugated-magnetic beads. The antibodies used in the immunoprecipitation of Flag tag, acetylated K of GKRP, and acetylated K of keratin 8 were anti-DYKDDDDK tag magnetic beads (1E6, 10 μl/mg protein; Wako), anti-AcK (#9441, 0.2 μl/mg protein; Cell Signaling Technology, Danvers, MA), and anti-AcK (ab21623, 0.2 μl/mg protein; Abcam, Cambridge, UK), respectively. The following antibodies were obtained for

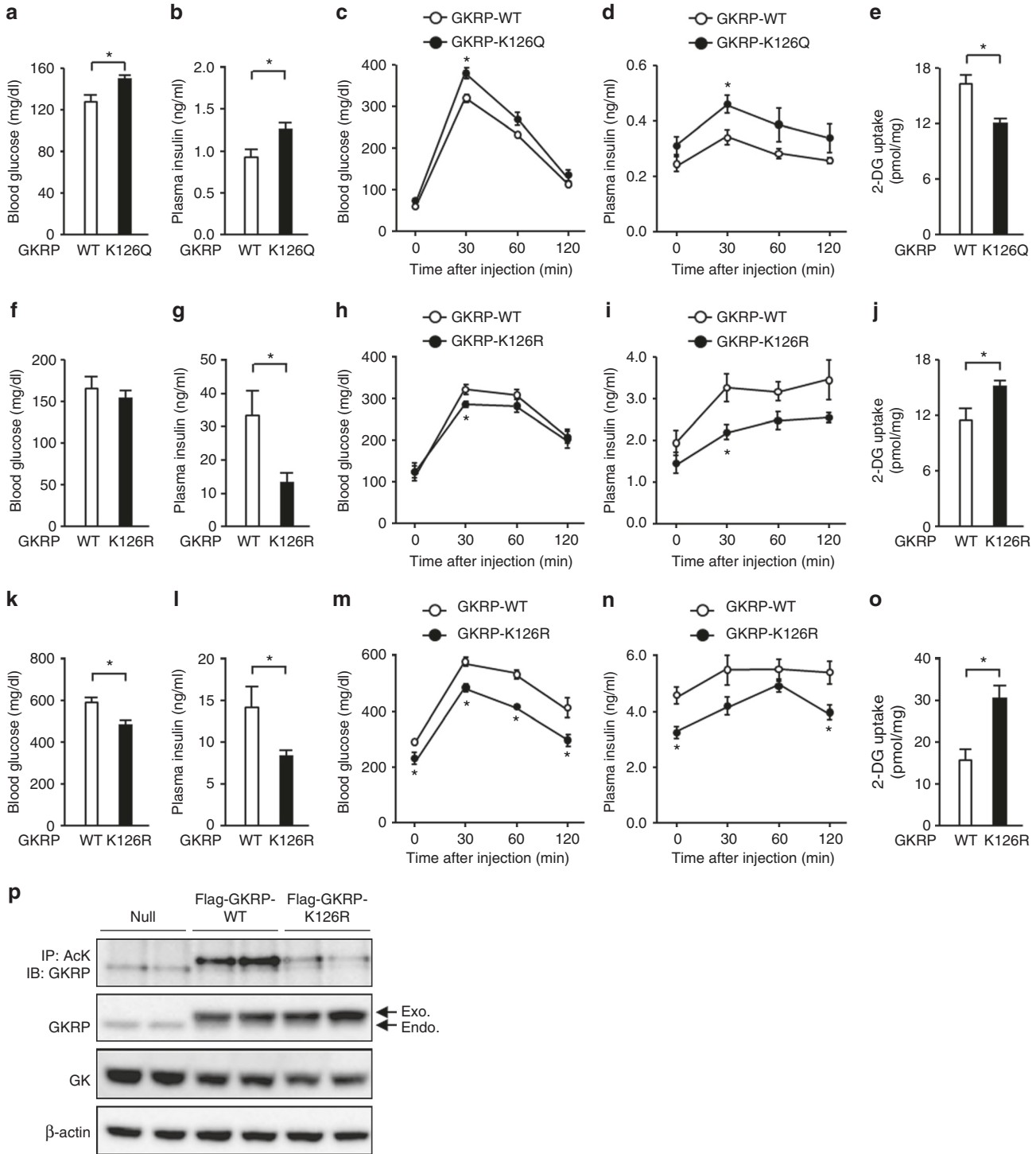

**Fig. 7** Deacetylation of K126 on GKRP improves hepatic glucose uptake and glucose intolerance in diabetic obese mice. **a–e** Effect of adenovirus-mediated hepatic overexpression of GKRP-K126Q or GKRP-wild-type (GKRP-WT) on blood glucose (**a**) and plasma insulin (**b**) levels under free-feeding conditions and changes in blood glucose (**c**), plasma insulin (**d**), and hepatic 2-deoxyglucose (2-DG) uptake (**e**) levels during glucose administration (2 g/kg) in lean mice ($n = 5$). **f–j** Effect of adenovirus-mediated hepatic overexpression of GKRP-K126R or GKRP-WT on blood glucose (**f**) and plasma insulin (**g**) levels under free-feeding conditions and changes in blood glucose (**h**), plasma insulin (**i**), and hepatic 2-DG uptake (**j**) levels after glucose administration (1 g/kg) in HFD mice ($n = 5$). **k–o** Effect of adenovirus-mediated hepatic overexpression of GKRP-K126R or GKRP-WT on blood glucose (**k**) and plasma insulin (**l**) levels under free-feeding conditions and changes in blood glucose (**m**), plasma insulin (**n**), and hepatic 2-DG uptake (**o**) levels after glucose administration (1 g/kg) in db/db mice ($n = 4$). **p** Acetylation level of GKRP in db/db mice transfected with adenovirus encoding Flag-GKRP-WT or Flag-GKRP-K126R. Endo. endogenous GKRP, Exo. exogenous GKRP. $*P < 0.05$; Student's $t$-test (**a–o**). Data in **p** are representative of at least three independent experiments. Error bars show s.e.m

immunoblotting: anti-Sirt2 (sc-20966, 1:500), anti-glucokinase (sc-7908, 1:500), anti-GKRP (sc-11416, 1:500) (Santa Cruz Biotechnology), anti-Sirt1 (07-131, 1:1000; Millipore, Billerica, MA), anti-Flag (F1804, 1:1000), anti-β-actin (A5441, 1:1000) (Sigma), anti-HA (3F10, 1:1000; Roche), anti-Halo (G9281, 1:1000; Promega), and anti-keratin 8 (TROMA-I, 1:200; Developmental Studies Hybridoma Bank, Iowa City, IA). Immunoblot images were quantified by densitometry on an LAS-3000 Imager (Fujifilm, Tokyo, Japan). Uncropped images of representative immunoblots are shown in Supplementary Fig. 9.

**Adenovirus vectors.** Adenovirus vectors (Flag-tagged Sirt2, HA-tagged Sirt2, Flag-tagged glucokinase, HA-tagged glucokinase, glucokinase, Flag-tagged GKRP, Flag-tagged K126R GKRP, Flag-tagged K126Q GKRP, and U6) were constructed by an Adenovirus Dual Expression Kit (Takara Bio). Adenovirus vector encoding Flag-tagged Sirt1 was provided by T Kitamura (Gunma University). Isolated mouse hepatocytes and mice were used for each experiment 24 h and 3–4 days after adenovirus transfection, respectively. Mice received adenoviruses ($5.0 \times 10^8$ plaque-forming units) into the tail vein.

**Protein−protein interaction assay.** The interaction between Sirt2 and GKRP or glucokinase was analyzed by a fluorescent-based system for detecting protein−protein interactions (MBL, Nagoya, Japan), which detects protein−protein interactions as fluorescent foci. Hepatocytes were transfected with phAG-tagged Sirt2, GKRP, or glucokinase and pAsh-tagged Sirt2, GKRP, or glucokinase vectors using Lipofectamine 3000 Reagent (Life Technologies). The intracellular fluorescent foci were observed using a fluorescence microscope (FSX100; Olympus, Tokyo, Japan).

**Mass spectrometry analysis of GKRP.** Whole-cell lysates of HEK293 cells transfected with Flag-tagged GKRP were immunoprecipitated with an anti-DYKDDDDK tag antibody (Wako). The fusion protein was eluted by DYKDDDDK Peptide (Wako) and re-immunoprecipitated by anti-AcK (Cell Signaling Technology). The immunoprecipitate was resolved on a 4–12% gradient SDS-PAGE gel and the protein band was subjected to mass spectrometry analysis by MS Bioworks (Ann Arbor, MI). For mass spectrometry analysis of the compounds in this article, see Supplementary Figure 5.

**Statistical analysis.** Data are presented as mean ± standard error of the mean. Statistical methods were not used to determine sample size before the experiment, which was instead based on preliminary data and previous publications. Animals were excluded from experiments if they showed any sign of sickness. Variance homogeneity was examined using Levene's test for multiple groups or an F-test for two groups. Statistical analysis was performed using the Student's $t$-test for the comparison of two groups and one-way ANOVA with the Fisher's PLSD post-hoc test for the comparison of several groups. Differences were considered significant at $P < 0.05$.

**Data availability.** The data that support the findings of this study are available from the corresponding authors upon reasonable request.

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

## Acknowledgements

We thank K. Nagamori, M. Nishio, and C. Asahi, Kanazawa University, for providing technical assistance, and ThinkSCIENCE (Tokyo, Japan) for help preparing the manuscript. This work was supported by a Grant-in-Aid for Scientific Research on Innovative Areas (Homeostatic regulation by various types of cell death) (17H05499) from the Ministry of Education, Culture, Sports, Science and Technology (MEXT), a Grant-in-Aid for Scientific Research (B) (26282022) and for Scientific Research (C) (16K00849) from the Japan Society for the Promotion of Science (JSPS), the Creation of Fundamental Technologies for Understanding and Control of Biosystem Dynamics (JPMJCR12W3), CREST from the Japan Science and Technology Agency (JST), and a research grant from the Astellas Foundation for Research on Metabolic Disorders, the Uehara Memorial Foundation, the Takeda Science Foundation, Novartis Pharma, and Eli Lilly Japan.

## Author contributions

H.W. obtained the data, contributed to the discussion, and wrote the manuscript. Y.I. and K.K. obtained the data. M.M., S.K., and M.K. contributed to the discussion and reviewed and edited the manuscript. H.I. researched the data, designed the study, contributed to the discussion, wrote the manuscript, and is the guarantor.

## Additional information

**Competing interests:** The authors declare no competing financial interests.

