## [Peer Review File · Nature Communications]

Reviewers' comments:

Reviewer #1 (Remarks to the Author):

Watanabe et al.

SIRT2 facilitates hepatic glucose uptake by deacetylating glucokinase regulatory protein

In this study, the authors showed lines of evidence supporting that the impairment of hepatic glucose uptake (HGU) in high-fat diet (HFD)-fed mice is caused by the reduction of SIRT2 activity, which increases the acetylation of glucokinase regulatory protein (GKRP) and promotes the dissociation of GKRP from glucokinase. This is likely due to the decrease in hepatic NAD⁺ levels, and therefore, restoring NAD⁺ levels by nicotinamide mononucleotide (NMN), a key NAD⁺ intermediate, enhances the deacetylation of GKRP and improves HGU. The authors used deacetylation- and acetylation-mimicking mutants of GKRP and nicely showed the importance of this SIRT2-dependent deacetylation of GKRP for the regulation of HGU.

This is a very interesting study, and the importance of SIRT2 for the regulation of GKRP function is novel. The weakness of this study, though, is the physiological significance of this regulation under a regular chow-fed (normal) condition. In particular, it still remains unclear how exactly the GKRP-GK association is regulated by changes in glucose levels (low glucose vs. high glucose). The authors should look a bit deeper into this aspect to strengthen the physiological significance of what they found in this study.

Major comments:

1) High glucose promotes the dissociation of GKRP from GK, and this dissociation is SIRT2-dependent (Figure 3a). However, the acetylation levels of GKRP do not seem to decrease in response to high glucose (Figures 4b and 4c). Yet, the acetylation status of K126 appears to be important for the SIRT2-dependent regulation of the GKRP-GK association (Figure 5d). These findings imply that deacetylation of K126 is indeed regulated by SIRT2, but this regulation is not as simple as the authors are claiming. The authors should further investigate how exactly the association of GKRP and GK is regulated in a glucose-dependent manner. For example, is the binding of SIRT2 to GKRP regulated in a glucose-dependent manner? In response to high glucose, is K126 deacetylated specifically on GKRP? Does NAD⁺ levels change in response to high glucose? What happens to HGU between low and high glucose conditions?

2) In IPGTTs with regular chow-fed mice, what happens to the acetylation levels of GKRP and the association of GKRP and GK? How about liver-specific SIRT2 KD mice? Do they show increases in GKRP acetylation levels and GKRP-GK association?

3) Under a HFD condition, the involvement of SIRT2 looks partial (Figure 2e). This result implies a possibility that other sirtuins, such as SIRT1, might also be involved in the regulation of HGU in this pathological condition. Sometimes, the functions of SIRT1 and SIRT2 are redundant, and therefore, single knockdown might not reveal the real importance of each of them. Therefore, it is important to conduct double knockdown of SIRT1 and SIRT2 and check both 2-DG uptake (Figure 2a) and IPGTTs in vivo (Figure 2e).

Minor comments:

1) Is it difficult to show the endogenous interaction between SIRT2 and GKRP?

2) Is SIRT1 also able to interact with GKRP?

3) The whole-body SIRT2 KO mice are viable. It might be interesting to check their livers.

Reviewer #2 (Remarks to the Author):

Low hepatic glucose uptake in models of T2D is frequently attributed to low glucokinase protein expression, based on evidence for low glucokinase enzyme activity in human T2D and in animal models. This study uses the db/db model and mice fed a high fat diet to study the role of GKR- acetylation status in hepatic glucose uptake. It argues based on the db/db model and mice fed a high fat diet that the defect in hepatic glucose uptake is due to the low hepatic low NAD content resulting in increased GKR acetylation which prevents dissociation of the GKR-GK complex. It demonstrates that the acetylation status of GKR protein is dependent on SIRT2 activity and on nutritional status but not on glucose concentration.

The MS reports that: (1) diminished SIRT2 activity impairs hepatic glucose uptake in db/db mice; (2) SIRT2 overexpression mitigates impaired HGU; (3) SIRT2 KD impairs GTT; (4) SIRT2 deacetylates GKR-K126; (5) Endogenous GK-GKR dissociation is impaired in db/db hepatocytes but a transfected GKR mutant that cannot be acetylated is not impaired. This is a thorough study that rigorously explores the role of SIRT2 and acetylation status of GKR.

This study has several major strengths of study in terms of mechanism: (i) It identifies K126 in GKR as the main Lysine residue involved in the GK-GKR interaction; (ii) it demonstrates by proximity ligation assays interaction of GKR and Sirt2 in the cytoplasm; (iii) it provides evidence for a role for SIRT2 in this mechanism.

Questions raised by this study are whether the 2DG assay is a robust assay of free GK activity (not bound to GKR) and whether the impaired hepatic glucose in db/db is exclusively due to the acetylation status of GKR.

Main points:

1. 2-Deoxyglucose assay: The study uses a 3 minute incubation with 2-DG followed by analysis of 2DG with an assay kit to measure glucose phosphorylation. The lower 2DG uptake in hepatocytes from db/db mice could be in part due to the lower GK protein shown in the Western Blot in Supplementary Fig. S1. However, the increase in 2DG by NMN is more likely to be explained by changes in covalent modification than by restoration of GK protein. It would be prudent to transfer the Western Blot image for GK-protein from FigS1 to Fig. 1. This image suggests to this reviewer that a role for low GK protein in this model of T2D cannot be firmly excluded (even though the effect of NMN is most likely on acetylation and not on GK protein).

2. The second sentence of the Discussion stating that "We found that HGU impairment in type 2 diabetes results from a decrease in hepatic NAD⁺ dependent SIRT2 activity.." etc should be modified to "We found that HGU impairment in the db/db model results in part from a decrease in hepatic NAD⁺ dependent SIRT2 activity.." Because it cannot be assumed that (1) Human T2D will be identical to the db/db model; (2) NMN administration will not have other effects.

RESPONSES TO REVIEWER COMMENTS

Manuscript: NCOMMS-17-09259, "SIRT2 facilitates hepatic glucose uptake by deacetylating glucokinase regulatory protein "

Reviewers' comments:

Reviewer #1 (Remarks to the Author):

Watanabe et al.

SIRT2 facilitates hepatic glucose uptake by deacetylating glucokinase regulatory protein

In this study, the authors showed lines of evidence supporting that the impairment of hepatic glucose uptake (HGU) in high-fat diet (HFD)-fed mice is caused by the reduction of SIRT2 activity, which increases the acetylation of glucokinase regulatory protein (GKRP) and promotes the dissociation of GKRP from glucokinase. This is likely due to the decrease in hepatic NAD⁺ levels, and therefore, restoring NAD⁺ levels by nicotinamide mononucleotide (NMN), a key NAD⁺ intermediate, enhances the deacetylation of GKRP and improves HGU. The authors used deacetylation- and acetylation-mimicking mutants of GKRP and nicely showed the importance of this SIRT2-dependent deacetylation of GKRP for the regulation of HGU.

This is a very interesting study, and the importance of SIRT2 for the regulation of GKRP function is novel. The weakness of this study, though, is the physiological significance of this regulation under a regular chow-fed (normal) condition. In particular, it still remains unclear how exactly the GKRP-GK association is regulated by changes in glucose levels (low glucose vs. high glucose). The authors should look a bit deeper into this aspect to strengthen the physiological significance of what they found in this study.

Major comments:

1) High glucose promotes the dissociation of GKRP from GK, and this dissociation is SIRT2-dependent (Figure 3a). However, the acetylation levels of GKRP do not seem to decrease in response to high glucose (Figures 4b and 4c). Yet, the acetylation status of K126 appears to be important for the SIRT2-dependent regulation of the GKRP-GK association (Figure 5d). These findings imply that deacetylation of K126 is indeed regulated by SIRT2, but this regulation is not as simple as the authors are claiming. The authors

should further investigate how exactly the association of GKR and GK is regulated in a glucose-dependent manner. For example, is the binding of SIRT2 to GKR regulated in a glucose-dependent manner? In response to high glucose, is K126 deacetylated specifically on GKR? Does NAD⁺ levels change in response to high glucose? What happens to HGU between low and high glucose conditions?

We have added the results of newly performed experiments in primary hepatocytes examining the effects of glucose on the SIRT2–GKR association to a new Fig. 3g, on deacetylation of GKR to a new Supplementary Fig. 4c, and on NAD⁺ levels to a new Supplementary Fig. 4b.

By using co-immunoprecipitation studies, we measured the SIRT2–GKR association under low- and high-glucose conditions and found that the SIRT2–GKR association remained unchanged even under high-glucose conditions (new Fig. 3g). We have added the following text to the Results: “The SIRT2–GKR co-immunoprecipitation remained unaltered by the glucose concentration in the culture medium in primary cultured hepatocytes, whereas the glucokinase–GKR co-immunoprecipitation was blocked under high-glucose conditions (Fig. 3g)”.

We measured the glucose-dependent change in GKR acetylation in primary hepatocytes in the original Fig. 4b. However, because the acetylation levels of GKR in lean mouse-derived hepatocytes may be too low to show their glucose-dependent change, we measured GKR acetylation and NAD⁺ levels in lean mouse-derived hepatocytes transduced with adenoviral vector encoding GKR. Overexpression of GKR made it easier to detect its acetylation and we found that both GKR acetylation and NAD⁺ levels were not affected by the glucose levels in the cell medium. We have added these results to a new Supplementary Fig. 4b and 4c and the following text to the Results: “GKR acetylation in primary hepatocytes was enhanced by SIRT2 knockdown but was unaffected by the glucose concentration in the culture medium, as with NAD⁺ levels, even in the hepatocytes overexpressing GKR (Fig. 4b and Supplementary Fig. 4b,c)”.

We apologize for failing to explain the mechanism of glucose-induced HGU, which has been reported previously. Glucose itself is reported to structurally rearrange glucokinase and induce glucokinase–GKR dissociation. We have added a new citation (Choi JM., et al., Proc Natl Acad Sci U S A., 2013) and the following explanation to the Discussion:

“Glucokinase is bound to GKR under low-glucose conditions, resulting in its inactivation. Glucose induces a structural rearrangement of glucokinase, which is followed by glucokinase–GKR dissociation and glucokinase activation²⁵. Indeed, co-immunoprecipitation of glucokinase and GKR decreased under high-glucose conditions in lean mouse-derived hepatocytes”.

Our new experiments revealed that GKR acetylation and SIRT2–GKR association were unaffected by glucose conditions, but we showed in the original manuscript that GKR acetylation was increased in db/db mouse-derived hepatocytes and impeded 2-deoxyglucose uptake. Therefore, we speculated that GKR acetylation is a checkpoint that allows hepatocyte glucose uptake depending on hepatocyte NAD⁺ levels, but not on glucose conditions. We have added the following to the Discussion to explain the results of the new experiments and our speculation: “We found that GKR deacetylation by SIRT2 is a check-point that approves their glucose-dependent dissociation, because pseudoacetylation GKR mutant (K126Q) could not dissociate from glucokinase under high-glucose conditions. However, as non-acetylatable GKR mutant (K126R) can bind to glucokinase under high-glucose conditions, deacetylation of GKR itself did not induce its dissociation from glucokinase. We also found that SIRT2 binds to GKR and that this association in lean mouse-derived hepatocytes was unaffected by glucose levels in the cell medium. These findings may indicate that SIRT2 binds to and deacetylates GKR to prevent GKR from binding to and inhibiting glucokinase in lean mouse liver, where SIRT2 activity is sufficient to deacetylate GKR even under high-glucose conditions; conversely, inactivated SIRT2 in obese-mouse liver fails to deacetylate GKR, resulting in GKR binding to and inhibiting glucokinase. Elucidation of how K126 acetylation prevents the glucose-dependent glucokinase–GKR dissociation, such as via structural changes in their complex, remains a future challenge.” We have also revised the subsequent paragraph in the Discussion to avoid content redundancy.

2) In IPGTTs with regular chow-fed mice, what happens to the acetylation levels of GKR and the association of GKR and GK? How about liver-specific SIRT2 KD mice? Do they show increases in GKR acetylation levels and GKR-GK association?

We measured the glucokinase–GKR association and GKR acetylation in IPGTT with regular chow-fed mice with and without liver SIRT2 knockdown and have added the results to a new Fig. 3d and new Fig. 4h, respectively. We have added the following to the Results: “Glucose administration induced glucokinase–GKR dissociation in the liver, but this dissociation was inhibited by hepatic SIRT2 knockdown (Fig. 3d)” and “The acetylation of both SIRT2 and keratin 8 was unaffected by glucose administration (Fig. 4h)”. To the Discussion, we have added the following: “GKR acetylation still showed little change after glucose administration and a minimal increase after refeeding in lean mice, but SIRT2 knockdown resulted in elevated GKR acetylation in lean mice. These findings reveal that SIRT2 activity suffices to deacetylate GKR in the livers of lean mice, even under fasting and refeeding conditions”. Due to the limited space, we transferred the original Fig. 4h to a

new Supplementary Fig. 4d, which shows a minimal increase in the hepatic acetylation of keratin 8 and GGRP after refeeding in lean mice.

3) Under a HFD condition, the involvement of SIRT2 looks partial (Figure 2e). This result implies a possibility that other sirtuins, such as SIRT1, might also be involved in the regulation of HGU in this pathological condition. Sometimes, the functions of SIRT1 and SIRT2 are redundant, and therefore, single knockdown might not reveal the real importance of each of them. Therefore, it is important to conduct double knockdown of SIRT1 and SIRT2 and check both 2-DG uptake (Figure 2a) and IPGTTs in vivo (Figure 2e).

Regarding the functional redundancy of SIRT2 and SIRT1, we have added a new citation (Haigis MC, Guarente LP., Genes Dev., 2006) and the following explanation to the Introduction: “The function of SIRT2 partially overlaps with that of SIRT1 because both deacetylate the histone H4K16”.

We showed that SIRT1 knockdown has an insignificant effect on 2-deoxyglucose uptake in primary hepatocytes in Fig. 2a. For further examination of the role of SIRT1 in the impeded hepatocyte glucose uptake, we measured 2-deoxyglucose uptake in db/db mouse-derived hepatocytes with SIRT1 overexpression and have added the results to a new Supplementary Fig. 3b and the following text to the Results: “SIRT2 overexpression increased 2-DG uptake by db/db mouse-derived primary hepatocytes, but SIRT1 overexpression did not... (Fig. 2c and Supplementary Fig. 3b)”. We have also added the following text to the Discussion: “While the function of SIRT1 partially overlaps that of SIRT2, neither SIRT1 knockdown in lean mouse-derived hepatocytes nor its overexpression in db/db mouse-derived hepatocytes affected 2-DG uptake”.

We also performed studies using double knockdown of SIRT1 and SIRT2 in primary hepatocytes and have added the results to a new Fig. 2d and new Fig. 2f and the following text to the results: “In spite of the functional redundancy between SIRT1 and SIRT2²¹, SIRT1 and SIRT2 double knockdown revealed no difference in 2-DG uptake compared with SIRT2 knockdown in lean mouse-derived hepatocytes or in NMN-treated db/db mouse-derived hepatocytes (Fig. 2d,f)”.

Further, we carried out IPGTT with continuous somatostatin administration in mice with hepatic SIRT1 and SIRT2 knockdown and have added the results on blood glucose levels to a new Fig. 2e and the plasma insulin levels to Supplementary Fig. 3c and the following text to the Results: “Furthermore, hepatic SIRT1 knockdown had no additional amelioration of the NMN-dependent increase in glucose tolerance compared with hepatic SIRT2 knockdown (Fig. 2e and Supplementary Fig. 3c)”.

Minor comments:

1) Is it difficult to show the endogenous interaction between SIRT2 and GKR?P?

We screened the immunoprecipitation utility of several commercial antibodies of SIRT2 and GKR?P but could not find a suitable antibody for immunoprecipitation. Hence, it was difficult to detect the endogenous interaction between SIRT2 and GKR?P. We are now trying to make a polyclonal antibody that can immunoprecipitate GKR?P, but some problems with antibody creation have been encountered. Detection of the endogenous interaction between SIRT2 and GKR?P remains a challenge for our laboratory.

2) Is SIRT1 also able to interact with GKR?P?

In new experiments using HEK293, SIRT2 bound to GKR?P, but SIRT1 did not. We have added the results to a new Supplementary Fig. 4a and the text to the Results: "In contrast, SIRT1 did not co-immunoprecipitate GKR?P (Supplementary Fig. 4a)". We have also added the following text to the Discussion: "Indeed, SIRT1 did not bind to GKR?P, whereas SIRT2 directly bound to GKR?P to regulate HGU".

3) The whole-body SIRT2 KO mice are viable. It might be interesting to check their livers.

We thank the reviewer for the suggestion. We used siRNA-mediated hepatic knockdown mice for this manuscript because of its specificity for the liver. However, we also believe that investigation of whole-body SIRT2 KO would be important to confirm the role of SIRT2 in HGU. However, because it takes more than 1 year to obtain KO mice, and longer for a sufficient number of experiments, we would like to investigate this aspect of SIRT2 as part of a future study.

Reviewer #2 (Remarks to the Author):

Low hepatic glucose uptake in models of T2D is frequently attributed to low glucokinase protein expression, based on evidence for low glucokinase enzyme activity in human T2D and in animal models. This study uses the db/db model and mice fed a high fat diet to study the role of GKR?P-acetylation status in hepatic glucose uptake. It argues based on the db/db model and mice fed a high fat diet that the defect in hepatic glucose uptake is due to the low hepatic low NAD content resulting in increased GKR?P acetylation which prevents dissociation of the GKR?P-GK complex. It demonstrates that the acetylation status of GKR?P protein is dependent on SIRT2 activity and on nutritional status but not on glucose concentration.

The MS reports that: (1) diminished SIRT2 activity impairs hepatic glucose uptake in db/db mice; (2) SIRT2 overexpression mitigates impaired HGU; (3) SIRT2 KD impairs GTT; (4) SIRT2 deacetylates GKRK-K126; (5) Endogenous GK-GKRK dissociation is impaired in db/db hepatocytes but a transfected GKRK mutant that cannot be acetylated is not impaired. This is a thorough study that rigorously explores the role of SIRT2 and acetylation status of GKRK.

This study has several major strengths of study in terms of mechanism: (i) It identifies K126 in GKRK as the main Lysine residue involved in the GK-GKRK interaction; (ii) it demonstrates by proximity ligation assays interaction of GKRK and Sirt2 in the cytoplasm; (iii) it provides evidence for a role for SIRT2 in this mechanism.

Questions raised by this study are whether the 2DG assay is a robust assay of free GK activity (not bound to GKRK) and whether the impaired hepatic glucose in db/db is exclusively due to the acetylation status of GKRK.

We examined whether 2-DG is taken up in the same manner as glucose and whether free glucokinase induces 2-DG uptake in GKRK knockdown hepatocytes. We found that 2-DG induced GK-GKRK dissociation and increased its uptake in the same manner as glucose (Fig. a and b, below). We also found that the 2-DG uptake increased in a dose-dependent manner with glucokinase expression in primary hepatocytes with GKRK knockdown (Fig. c, below). Further, 2-DG uptake decreased in a dose-dependent manner with GKRK expression in primary GKRK knockdown hepatocytes with forced glucokinase expression (Fig. c, below). We have not added these results to the manuscript because they are not directly related to the main topic of this manuscript. However, we will add these results as supplementary information if requested.

We assume that the pathological conditions of the impaired HGU in db/db mice change with age and growing environment and that the impaired HGU in db/db mice involves a number of factors, including increased gluconeogenesis and decreased glucokinase protein expression. In this manuscript, we clarified that impeded NAD⁺/SIRT2-dependent HGU regulation plays a significant role in the pathogenesis of impaired HGU in db/db mice. As written in the response to main point #2, we rephrased the indicated sentence in the Discussion to, “We found that HGU impairment in obese diabetic mice results in part from a decrease in hepatic NAD⁺-dependent SIRT2 activity...”.

Main points:

1. 2-Deoxyglucose assay: The study uses a 3 minute incubation with 2-DG followed by analysis of 2DG with an assay kit to measure glucose phosphorylation. The lower 2DG uptake in hepatocytes from db/db mice could be in part due to the lower GK protein shown in the Western Blot in Supplementary Fig. S1. However, the increase in 2DG by NMN is more

likely to be explained by changes in covalent modification than by restoration of GK protein. It would be prudent to transfer the Western Blot image for GK-protein from FigS1 to Fig. 1. This image suggests to this reviewer that a role for low GK protein in this model of T2D cannot be firmly excluded (even though the effect of NMN is most likely on acetylation and not on GK protein).

We agree with the reviewer's understanding of the original Supplementary Fig. S1b. In accordance with the suggestion, we have transferred the right panel of the original Supplementary Fig. S1b to a new Fig.1d.

2. The second sentence of the Discussion stating that "We found that HGU impairment in type 2 diabetes results from a decrease in hepatic NAD⁺ dependent SIRT2 activity.." etc should be modified to "We found that HGU impairment in the db/db model results in part from a decrease in hepatic NAD⁺ dependent SIRT2 activity.." Because it cannot be assumed that (1) Human T2D will be identical to the db/db model; (2) NMN administration will not have other effects.

We agree with the suggestion of the reviewer and have rephrased the sentence to "We found that HGU impairment in obese diabetic mice results in part from a decrease in hepatic NAD⁺-dependent SIRT2 activity", because we used the db/db model and mice fed a high-fat diet.

We also rephrased "we found that NAD⁺ depletion is involved in the HGU impairment of type 2 diabetes..." in the Introduction to "we found that NAD⁺ depletion is involved in the HGU impairment of obese diabetic mice...", for the same reason the reviewer indicated.

REVIEWERS' COMMENTS:

Reviewer #1 (Remarks to the Author):

Watanabe et al.

SIRT2 facilitates hepatic glucose uptake by deacetylating glucokinase regulatory protein

In this revised manuscript, the authors addressed reviewers' comments as much as they could. One important point the authors clarified with additional results is that SIRT2-dependent regulation of GK-GKRP interaction is totally separated from glucose-dependent regulation of their interaction. The authors did a pretty good job to clearly demonstrate this point.

Then, the important question, namely, what the physiological relevance of this SIRT2-dependent GK-GKRP regulation is, still remains. The authors argues in the Discussion section that "GKRP deacetylation by SIRT2 is a check-point that approves their glucose-dependent dissociation." But this interpretation could be still a bit problematic because this pulls a question of what triggers SIRT2's check-point or dissociation-approving function. Based on the data they show, SIRT2 appears to function as a NAD⁺-dependent rheostat that determines the extent of the interaction between GK and GKRP under a high glucose condition. In other words, SIRT2 always interacts with GKRP and simply determines the strength of interaction between GK and GKRP, following NAD⁺ levels. This implicates that SIRT2 function does not matter much when NAD⁺ levels are above its K_m (physiologically normal condition), but SIRT2's role becomes important when NAD⁺ levels decrease below SIRT2's K_m for NAD⁺, for example, under a high-fat diet condition.

Thus, the authors had better revise their discussion carefully and further clarify the role of SIRT2 in terms of the regulation of GK-GKRP interaction.

Reviewer #2 (Remarks to the Author):

This MS reports a novel mechanism whereby acetylation of GKRP in models of dietary obesity and hyperphagia contributes to increased sequestration of glucokinase in the nucleus and compromised dissociation by elevated glucose. This reviewer recommends that 3 statements in the Discussion that are unclear and may need clarification:

(1) The sentence in lines 333 to 336: This reviewer assumes that the authors mean: "it is plausible that a decrease in SIRT2 activity in the obese diabetic livers results in compromised regulation of HGU and glucokinase translocation by postprandial hyperglycemia"

(2) Line 343: "may be in part responsible"

(3) Line 386 "inactivated SIRT2" presumably "compromised SIRT2 activity" because this may be due to the low NAD rather than covalent inactivation.

RESPONSES TO REVIEWER COMMENTS

Manuscript: NCOMMS-17-09259A, "SIRT2 facilitates hepatic glucose uptake by deacetylating glucokinase regulatory protein"

REVIEWERS' COMMENTS:

Reviewer #1 (Remarks to the Author):

Watanabe et al.

SIRT2 facilitates hepatic glucose uptake by deacetylating glucokinase regulatory protein

In this revised manuscript, the authors addressed reviewers' comments as much as they could. One important point the authors clarified with additional results is that SIRT2-dependent regulation of GK-GKRP interaction is totally separated from glucose-dependent regulation of their interaction. The authors did a pretty good job to clearly demonstrate this point.

Then, the important question, namely, what the physiological relevance of this SIRT2-dependent GK-GKRP regulation is, still remains. The authors argues in the Discussion section that "GKRP deacetylation by SIRT2 is a check-point that approves their glucose-dependent dissociation." But this interpretation could be still a bit problematic because this pulls a question of what triggers SIRT2's check-point or dissociation-approving function. Based on the data they show, SIRT2 appears to function as a NAD⁺-dependent rheostat that determines the extent of the interaction between GK and GKRP under a high glucose condition. In other words, SIRT2 always interacts with GKRP and simply determines the strength of interaction between GK and GKRP, following NAD⁺ levels. This implicates that SIRT2 function does not matter much when NAD⁺ levels are above its K_m (physiologically normal condition), but SIRT2's role becomes important when NAD⁺ levels decrease below SIRT2's K_m for NAD⁺, for example, under a high-fat diet condition.

Thus, the authors had better revise their discussion carefully and further clarify the role of SIRT2 in terms of the regulation of GK-GKRP interaction.

Response: We agree with the reviewer's suggestion. We have revised the following sentences in the Discussion: "Sirt2 appears to function as a NAD⁺-dependent rheostat that

determines the extent of the interaction between GK and GKRP via GKRP deacetylation under a high-glucose condition. Actually, acetylation-mimicking GKRP mutant (K126Q) could not dissociate from glucokinase under high-glucose conditions. However, as deacetylation-mimicking GKRP mutant (K126R) can bind to glucokinase under high-glucose conditions, deacetylation of GKRP itself did not induce its dissociation from glucokinase. We also found that Sirt2 binds to GKRP and that this association in lean mouse-derived hepatocytes was unaffected by glucose levels in the cell medium. These findings may indicate that Sirt2 binds to and deacetylates GKRP to prevent GKRP from binding to and inhibiting glucokinase in lean mouse liver, where NAD⁺ levels are sufficient for Sirt2 to deacetylate GKRP even under high-glucose conditions; conversely, compromised Sirt2 activity caused by NAD⁺ reduction in obese-mouse liver fails to deacetylate GKRP, resulting in GKRP binding to and inhibiting glucokinase.”

Reviewer #2 (Remarks to the Author):

This MS reports a novel mechanism whereby acetylation of GKRP in models of dietary obesity and hyperphagia contributes to increased sequestration of glucokinase in the nucleus and compromised dissociation by elevated glucose. This reviewer recommends that 3 statements in the Discussion that are unclear and may need clarification:

(1) The sentence in lines 333 to 336: This reviewer assumes that the authors mean: "it is plausible that a decrease in SIRT2 activity in the obese diabetic livers results in compromised regulation of HGU and glucokinase translocation by postprandial hyperglycemia"

Response: We agree with the reviewer’s suggestion and have rephrased the sentence to “...it is plausible that a decrease in Sirt2 activity in the obese diabetic livers results in compromised regulation of HGU and glucokinase translocation by postprandial hyperglycemia”.

(2) Line 343: "may be in part responsible"

Response: We have added the words “in part”.

(3) Line 386 "inactivated SIRT2" presumably "compromised SIRT2 activity" because this may be due to the low NAD rather than covalent inactivation.

Response: We agree that Sirt2 activity is determined by the NAD⁺ level and have thus changed “inactivated SIRT2” to “compromised Sirt2 activity”.